# Satisfying Real-world Goals with Dataset Constraints

**Gabriel Goh**
Dept. of Mathematics
UC Davis
Davis, CA 95616
ggoh@math.ucdavis.edu

**Andrew Cotter, Maya Gupta**
Google Inc.
1600 Amphitheatre Parkway
Mountain View, CA 94043
acotter@google.com
mayagupta@google.com

**Michael Friedlander**
Dept. of Computer Science
University of British Columbia
Vancouver, B.C. V6T 1Z4
mpf@cs.ubc.ca

## Abstract

The goal of minimizing misclassification error on a training set is often just one of several real-world goals that might be defined on different datasets. For example, one may require a classifier to also make positive predictions at some specified rate for some subpopulation (fairness), or to achieve a specified empirical recall. Other real-world goals include reducing churn with respect to a previously deployed model, or stabilizing online training. In this paper we propose handling multiple goals on multiple datasets by training with dataset constraints, using the ramp penalty to accurately quantify costs, and present an efficient algorithm to approximately optimize the resulting non-convex constrained optimization problem. Experiments on both benchmark and real-world industry datasets demonstrate the effectiveness of our approach.

## 1   Real-world goals

We consider a broad set of design goals important for making classifiers work well in real-world applications, and discuss how metrics quantifying many of these goals can be represented in a particular optimization framework. The key theme is that these metrics, which range from the standard precision and recall, to less well-known examples such as coverage and fairness [17, 27, 15], and including some new proposals, can be expressed in terms of the positive and negative classification rates on multiple datasets.

**Coverage:**  One may wish to control how often a classifier predicts the positive (or negative) class. For example, one may want to ensure that only $10\%$ of customers are selected to receive a printed catalog due to budget constraints, or perhaps to compensate for a biased training set. In practice, constraining the "coverage rate" (the expected proportion of positive predictions) is often easier than measuring e.g. accuracy or precision because coverage can be computed on unlabeled data—labeling data can be expensive, but acquiring a large number of unlabeled examples is often very easy.

Coverage was also considered by Mann and McCallum [17], who proposed what they call "label regularization", in which one adds a regularizer penalizing the relative entropy between the mean score for each class and the desired distribution, with an additional correction to avoid degeneracies.

**Churn:**  Work does not stop once a machine learning model has been adopted. There will be new training data, improved features, and potentially new model structures. Hence, in practice, one will deploy a *series* of models, each improving slightly upon the last. In this setting, determining whether each candidate should be deployed is surprisingly challenging: if we evaluate on the *same* held-out testing set every time a new candidate is proposed, and deploy it if it outperforms its predecessor, then every compare-and-deploy decision will increase the statistical dependence between the deployed model and the testing dataset, causing the model sequence to fit the originally-independent testing data. This problem is magnified if, as is typical, the candidate models tend to disagree only on a relatively small number of examples near the true decision boundary.

A simple and safe solution is to draw a *fresh* testing sample every time one wishes to compare two models in the sequence, only considering examples on which the two models disagree. Because labeling data is expensive, one would like these freshly sampled testing datasets to be as small as possible. It is here that the problem of "churn" arises. Imagine that model A, our deployed model, is 70% accurate, and that model B, our candidate, is 75% accurate. In the best case, only 5% of test samples would be labeled differently, and all differences would be "wins" for classifier B. Then only a dozen or so examples would need to be labeled in order to establish that B is the statistically significantly better classifier with 95% confidence. In the worst case, model A would be correct and model B incorrect 25% of the time, model B correct and model A incorrect 30% of the time, and both models correct the remaining 45% of the time. Then 55% of testing examples will be labeled differently, and closer to 1000 examples would need to be labeled to determine that model B is better.

We define the "churn rate" as the expected proportion of examples on which the prediction of the model being considered (model B above) differs from that of the currently-deployed model (model A). During training, we propose constraining the empirical churn rate with respect to a given deployed model on a large unlabeled dataset (see also Fard et al. [12] for an alternative approach).

**Stability:** A special case of minimizing churn is to ensure stability of an online classifier as it evolves, by constraining it to not deviate too far from a trusted classifier on a large held-out unlabeled dataset.

**Fairness:** A practitioner may be required to guarantee *fairness* of a learned classifier, in the sense that it makes positive predictions on different subgroups at certain rates. For example, one might require that housing loans be given equally to people of different genders. Hardt et al. [15] identify three types of fairness: (i) demographic parity, in which positive predictions are made at the same rate on each subgroup, (ii) equal opportunity, in which only the true positive rates must match, and (iii) equalized odds, in which both the true positive rates and false positive rates must match. Fairness can also be specified by a proportion, such as the 80% rule in US law that certain decisions must be in favor of group B individuals at least 80% as often as group A individuals [e.g. 3, 26, 27, 15].

Zafar et al. [27] propose learning fair classifiers by imposing linear constraints on the covariance between the predicted labels and the values of certain features, while Hardt et al. [15] propose first learning an "unfair" classifier, and then choosing population-dependent thresholds to satisfy the desired fairness criterion. In our framework, rate constraints such as those mentioned above can be imposed directly, at training time.

**Recall and Precision:** Requirements of real-world classifiers are often expressed in terms of precision and recall, especially when examples are highly imbalanced between positives and negatives. In our framework, we can handle this problem via Neyman-Pearson classification [e.g. 23, 9], in which one seeks to minimize the false negative rate subject to a constraint on the false positive rate. Indeed, our ramp-loss formulation is equivalent to that of Gasso et al. [13] in this setting.

**Egregious Examples:** For certain classification applications, examples may be discovered that are particularly embarrassing if classified incorrectly. One standard approach to handling such examples is to increase their weights during training, but this is difficult to get right: too large a weight may distort the classifier too much in the surrounding feature space, whereas too small a weight may not fix the problem. Worse, over time the dataset will often be augmented with new training examples and new features, causing the ideal weights to drift. We propose instead simply adding a constraint ensuring that some proportion of a set of such egregious examples is correctly classified. Such constraints should be used with extreme care, since they can cause the problem to become infeasible.

## 2   Optimization problem

A key aspect of many of the goals of Section 1 is that they are defined on different datasets. For example, we might seek to maximize the accuracy on a set of labeled examples drawn in some biased manner, require that its recall be at least 90% on 50 small datasets sampled in an unbiased manner from 50 different countries, desire low churn relative to a deployed classifier on a large unbiased unlabeled dataset, and require that 100 given egregious examples be classified correctly.

Another characteristic common to the metrics of Section 1 is that they can be expressed in terms of the positive and negative classification rates on various datasets. We consider only *unlabeled* datasets, as described in Table 1—a dataset with binary labels, for example, would be handled by partitioning it into the two unlabeled datasets $D^+$ and $D^-$ containing the positive and negative examples,

Table 1: Dataset notation.

| Notation | Dataset |
|---|---|
| $D$ | Any dataset |
| $D^+, D^-$ | Sets of examples labeled positive/negative, respectively |
| $D^{++}, D^{+-}, D^{-+}, D^{--}$ | Sets of examples with ground-truth positive/negative labels, and for which a baseline classifier makes positive/negative predictions |
| $D^A, D^B$ | Sets of examples belonging to subpopulation A and B, respectively |

Table 2: The quantities discussed in Section 1, expressed in the notation used in Problem 1, with the dependence on $w$ and $b$ dropped for notational simplicity, and using the dataset notation of Table 1.

| Metric | Expression |
|---|---|
| Coverage rate | $s_p(D)$ |
| #TP, #TN, #FP, #FN | $|D^+| s_p(D^+), |D^-| s_n(D^-), |D^-| s_p(D^-), |D^+| s_n(D^+)$ |
| #Errors | #FP + #FN |
| Error rate | #Errors$/(|D^+| + |D^-|)$ |
| Recall | #TP$/$(#TP + #FN) = #TP$/|D^+|$ |
| #Changes | $|D^{+-}| s_p(D^{+-}) + |D^{-+}| s_n(D^{-+}) + |D^{+-}| s_p(D^{+-}) + |D^{-+}| s_n(D^{-+})$ |
| Churn rate | #Changes$/(|D^{++}| + |D^{+-}| + |D^{-+}| + |D^{--}|)$ |
| Fairness constraint | $s_p(D^A) \geq \kappa s_p(D^B)$, where $\kappa > 0$ |
| Equal opportunity constraint | $s_p(D^A \cap D^+) \geq \kappa s_p(D^B \cap D^+)$, where $\kappa > 0$ |
| Egregious example constraint | $s_p(D^+) \geq \kappa$ and/or $s_n(D^-) \leq \kappa$ for a dataset $D$ of egregious examples, where $\kappa \in [0,1]$ |

respectively. We wish to learn a linear classification function $f(x) = \langle w, x \rangle - b$ parameterized by a weight vector $w \in \mathbb{R}^d$ and bias $b \in \mathbb{R}$, for which the positive and negative classification rates are:

$$s_p(D; w, b) = \tfrac{1}{|D|} \sum_{x \in D} \mathbf{1}(\langle w, x \rangle - b), \qquad s_n(D; w, b) = s_p(D; -w, -b), \qquad (1)$$

where $\mathbf{1}$ is an indicator function that is 1 if its argument is positive, 0 otherwise. In words, $s_p(D; w, b)$ and $s_n(D; w, b)$ denote the proportion of positive or negative predictions, respectively, that $f$ makes on $D$. Table 2 specifies how the metrics of Section 1 can be expressed in terms of the $s_p$s and $s_n$s.

We propose handling these goals by minimizing an $\ell^2$-regularized positive linear combination of prediction rates on different datasets, subject to upper-bound constraints on other positive linear combinations of such prediction rates:

**Problem 1.** *Starting point: discontinuous constrained problem*

$$\begin{aligned}
\underset{w \in \mathbb{R}^d, b \in \mathbb{R}}{\text{minimize}} \quad & \sum_{i=1}^{k} \left( \alpha_i^{(0)} s_p(D_i; w, b) + \beta_i^{(0)} s_n(D_i; w, b) \right) + \tfrac{\lambda}{2} \|w\|_2^2 \\
\text{s.t.} \quad & \sum_{i=1}^{k} \left( \alpha_i^{(j)} s_p(D_i; w, b) + \beta_i^{(j)} s_n(D_i; w, b) \right) \leq \gamma^{(j)} \quad j \in \{1, \ldots, m\}.
\end{aligned}$$

Here, $\lambda$ is the parameter on the $\ell^2$ regularizer, there are $k$ unlabeled datasets $D_1, \ldots, D_k$ and $m$ constraints. The metrics minimized by the objective and bounded by the constraints are specified via the choices of the nonnegative coefficients $\alpha_i^{(0)}, \beta_i^{(0)}, \alpha_i^{(j)}, \beta_i^{(j)}$ and upper bounds $\gamma^{(j)}$ for the $i$th dataset and, where applicable, the $j$th constraint—a user should base these choices on Table 2. Note that because $s_p + s_n = 1$, it is possible to transform *any* linear combination of rates into an equivalent positive linear combination, plus a constant (see Appendix B[1] for an example).

We cannot optimize Problem 1 directly because the rate functions $s_p$ and $s_n$ are discontinuous. We can, however, work around this difficulty by training a classifier that makes *randomized* predictions based on the ramp function [7]:

$$\sigma(z) = \max\{0, \min\{1, 1/2 + z\}\}, \qquad (2)$$

**Algorithm 1** Proposed majorization-minimization procedure for (approximately) optimizing Problem 2. Starting from an initial feasible solution $w^{(0)}, b_0$, we repeatedly find a convex upper bound problem that is tight at the current candidate solution, and optimize it to yield the next candidate. See Section 2.1 for details, and Section 2.2 for how one can perform the inner optimizations on line 3.

> MajorizationMinimization $\left(w^{(0)}, b_0, T\right)$
> **1**    For $t \in \{1, 2, \dots, T\}$
> **2**        Construct an instance of Problem 3 with $w' = w^{(t-1)}$ and $b' = b_{t-1}$
> **3**        Optimize this convex optimization problem to yield $w^{(t)}$ and $b_t$
> **4**    Return $w^{(t)}, b_t$

where the randomized classifier parameterized by $w$ and $b$ will make a positive prediction on $x$ with probability $\sigma\left(\langle w, x\rangle - b\right)$, and a negative prediction otherwise (see Appendix A for more on this randomized classification rule). For this randomized classifier, the *expected* positive and negative rates will be:

$$r_p\left(D; w, b\right) = \tfrac{1}{|D|}\sum_{x \in D}\sigma\left(\langle w, x\rangle - b\right), \qquad r_n\left(D; w, b\right) = r_p\left(D; -w, -b\right). \qquad (3)$$

Using these expected rates yields a continuous (but non-convex) analogue of Problem 1:

**Problem 2.** *Ramp version of Problem 1*

$$\min_{w \in \mathbb{R}^d, b \in \mathbb{R}}\; \sum_{i=1}^{k}\left(\alpha_i^{(0)}r_p(D_i; w, b) + \beta_i^{(0)}r_n(D_i; w, b)\right) + \tfrac{\lambda}{2}\|w\|_2^2$$

$$\text{s.t.}\; \sum_{i=1}^{k}\left(\alpha_i^{(j)}r_p(D_i; w, b) + \beta_i^{(j)}r_n(D_i; w, b)\right) \leq \gamma^{(j)} \quad j \in \{1, \dots, m\}.$$

Efficient optimization of this problem is the ultimate goal of this section. In Section 2.1, we will propose a majorization-minimization approach that sequentially minimizes convex upper bounds on Problem 2, and, in Section 2.2, will discuss how these convex upper bounds may themselves be efficiently optimized.

## 2.1    Optimizing the ramp problem

To address the non-convexity of Problem 2, we will iteratively optimize approximations, by, starting from an feasible initial candidate solution, constructing a convex optimization problem upper-bounding Problem 2 that is *tight* at the current candidate, optimizing this convex problem to yield the next candidate, and repeating.

Our choice of a ramp for $\sigma$ makes finding such tight convex upper bounds easy: both the hinge function $\max\{0, 1/2 + z\}$ and constant-1 function are upper bounds on $\sigma$, with the former being tight for all $z \leq 1/2$, and the latter for all $z \geq 1/2$ (see Figure 1). We'll therefore define the following upper bounds on $\sigma$ and $1 - \sigma$, with the additional parameter $z'$ determining which of the two bounds (hinge or constant) will be used, such that the bounds will always be tight for $z = z'$:

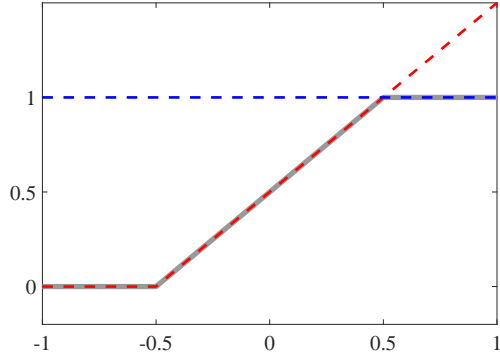

Figure 1: Convex upper bounds on the ramp function $\sigma(z) = \max\{0, \min\{1, 1/2 + z\}\}$. Notice that the hinge bound (red) is tight for all $z \leq 1/2$, and the constant bound (blue) is tight for all $z \geq 1/2$.

$$\check{\sigma}_p\left(z; z'\right) = \begin{cases} \max\{0, 1/2 + z\} & \text{if } z' \leq 1/2 \\ 1 & \text{otherwise} \end{cases}, \qquad \check{\sigma}_n(z; z') = \check{\sigma}_p\left(-z; -z'\right). \qquad (4)$$

Based upon these we define the following upper bounds on the expected rates:

$$\check{r}_p\left(D; w, b; w', b'\right) = \tfrac{1}{|D|}\sum_{x \in D}\check{\sigma}_p\left(\langle w, x\rangle - b; \langle w', x\rangle - b'\right) \qquad (5)$$

$$\check{r}_n\left(D; w, b; w', b'\right) = \tfrac{1}{|D|}\sum_{x \in D}\check{\sigma}_n\left(\langle w, x\rangle - b; \langle w', x\rangle - b'\right),$$

which have the properties that both $\check{r}_p$ and $\check{r}_n$ are convex in $w$ and $b$, are upper bounds on the original ramp-based rates:

$$\check{r}_p\left(D; w, b; w', b'\right) \geq r_p\left(D; w, b\right) \quad \text{and} \quad \check{r}_n\left(D; w, b; w', b'\right) \geq r_n\left(D; w, b\right),$$

**Algorithm 2** Skeleton of a cutting-plane algorithm that optimizes Equation 6 to within $\epsilon$ for $v \in \mathcal{V}$, where $\mathcal{V} \subseteq \mathbb{R}^m$ is compact and convex. Here, $l_0, u_0 \in \mathbb{R}$ are finite with $l_0 \leq \max_{v \in \mathcal{V}} F(v) \leq u_0$. There are several options for the CutChooser function on line 8—please see Appendix E for details. The SVMOptimizer function returns $w^{(t)}$ and $b_t$ approximately minimizing $\Psi(w, b, v^{(t)}; w', b')$, and a lower bound $l_t \leq F(v)$ for which $u_t - l_t \leq \epsilon_t$ for $u_t$ as defined on line 10.

---

CuttingPlane $(l_0, u_0, \mathcal{V}, \epsilon)$

**1**     Initialize $g^{(0)} \in \mathbb{R}^m$ to the all-zero vector

**2**     For $t \in \{1, 2, \dots\}$

**3**        Let $h_t(v) = \min_{s \in \{0,1,\dots,t-1\}} \left( u_s + \langle g^{(s)}, v - v^{(s)} \rangle \right)$

**4**        Let $L_t = \max_{s \in \{0,1,\dots,t-1\}} l_s$ and $U_t = \max_{v \in \mathcal{V}} h_t(v)$

**5**        If $U_t - L_t \leq \epsilon$ then

**6**           Let $s \in \{1, \dots, t-1\}$ be an index maximizing $l_s$

**7**           Return $w^{(s)}, b_s, v^{(s)}$

**8**        Let $v^{(t)}, \epsilon_t = \text{CutChooser}(h_t, L_t)$

**9**        Let $w^{(t)}, b_t, l_t = \text{SVMOptimizer}\left( v^{(t)}, h_t\left(v^{(t)}\right), \epsilon_t \right)$

**10**      Let $u_t = \Psi(w^{(t)}, b_t, v^{(t)}; w', b')$ and $g^{(t)} = \nabla_v \Psi(w^{(t)}, b_t, v^{(t)}; w', b')$

---

and are tight at $w', b'$:

$$\check{r}_p(D; w', b'; w', b') = r_p(D; w', b') \quad \text{and} \quad \check{r}_n(D; w', b'; w', b') = r_n(D; w', b').$$

Substituting these bounds into Problem 2 yields:

**Problem 3.** *Convex upper bound on Problem 2*

$$\operatorname*{minimize}_{w \in \mathbb{R}^d, b \in \mathbb{R}} \quad \sum_{i=1}^{k} \left( \alpha_i^{(0)} \check{r}_p(D_i; w, b; w', b') + \beta_i^{(0)} \check{r}_n(D_i; w, b; w', b') \right) + \frac{\lambda}{2} \|w\|_2^2$$

$$\text{s.t.} \quad \sum_{i=1}^{k} \left( \alpha_i^{(j)} \check{r}_p(D_i; w, b; w', b') + \beta_i^{(j)} \check{r}_n(D_i; w, b; w', b') \right) \leq \gamma^{(j)} \quad j \in \{1, \dots, m\}.$$

As desired, this problem upper bounds Problem 2, is tight at $w', b'$, and is convex (because any positive linear combination of convex functions is convex).

Algorithm 1 contains our proposed procedure for approximately solving Problem 2. Given an initial feasible solution, it's straightforward to verify inductively, using the fact that we construct tight convex upper bounds at every step, that every convex subproblem will have a feasible solution, every $(w^{(t)}, b_t)$ pair will be feasible w.r.t. Problem 2, and every $(w^{(t+1)}, b_{t+1})$ will have an objective function value that is no larger that that of $(w^{(t)}, b_t)$. In other words, no iteration can make negative progress. The non-convexity of Problem 2, however, will cause Algorithm 1 to arrive at a suboptimal solution that depends on the initial $(w^{(0)}, b_0)$.

### 2.2 Optimizing the convex subproblems

The first step in optimizing Problem 3 is to add Lagrange multipliers $v$ over the constraints, yielding the equivalent unconstrained problem:

$$\operatorname*{maximize}_{v \succeq 0} \ F(v) = \min_{w, b} \Psi(w, b, v; w', b'), \tag{6}$$

where the function:

$$\Psi(w, b, v; w', b') = \sum_{i=1}^{k} \left( \left( \alpha_i^{(0)} + \sum_{j=1}^{m} v_j \alpha_i^{(j)} \right) \check{r}_p(D_i; w, b; w', b') \right. \tag{7}$$

$$\left. + \left( \beta_i^{(0)} + \sum_{j=1}^{m} v_j \beta_i^{(j)} \right) \check{r}_n(D_i; w, b; w', b') \right) + \frac{\lambda}{2} \|w\|_2^2 - \sum_{j=1}^{m} v_j \gamma^{(j)}$$

is convex in $w$ and $b$, and concave in the multipliers $v$. For the purposes of this section, $w'$ and $b'$, which were found in the previous iteration of Algorithm 1, are fixed constants.

Because this is a convex-concave saddle point problem, there are a large number of optimization techniques that could be successfully applied. For example, in settings similar to our own, Eban et al. [10] simply perform SGD jointly over all parameters (including $v$), while Gasso et al. [13] use the Uzawa algorithm, which would alternate between (i) optimizing exactly over $w$ and $b$, and (ii) taking gradient steps on $v$.

We instead propose an approach for which, in our setting, it is particularly easy to create an efficient implementation. The key insight is that evaluating $F(v)$ is, thanks to our use of hinge and constant upper-bounds on our ramp $\sigma$, equivalent to optimization of a support vector machine (SVM) with per-example weights—see Appendix F for details. This observation enables us to solve the saddle system in an inside-out manner. On the "inside", we optimize over $(w, b)$ for fixed $v$ using an off-the-shelf SVM solver [e.g. 6]. On the "outside", the resulting $(w, b)$-optimizer is used as a component in a cutting-plane optimization over $v$. Notice that this outer optimization is very low-dimensional, since $v \in \mathbb{R}^m$, where $m$ is the number of constraints.

Algorithm 2 contains a skeleton of the cutting-plane algorithm that we use for this outer optimization over $v$. Because this algorithm is intended to be used as an outer loop in a nested optimization routine, it does not expect that $F(v)$ can be evaluated or differentiated exactly. Rather, it's based upon the idea of possibly making "shallow" cuts [4] by choosing a desired accuracy $\epsilon_t$ at each iteration, and expecting the SVMOptimizer to return a solution with suboptimality $\epsilon_t$. More precisely, the SVMOptimizer function approximately evaluates $F(v^{(t)})$ for a given fixed $v^{(t)}$ by constructing the corresponding SVM problem and finding a $(w^{(t)}, b_t)$ for which the primal and dual objective function values differ by at most $\epsilon_t$.

After finding $(w^{(t)}, b_t)$, the SVMOptimizer then evaluates the dual objective function value of the SVM to determine $l_t$. The primal objective function value $u_t$ and its gradient $g^{(t)}$ w.r.t. $v$ (calculated on line 10 of Algorithm 2) define the cut $u_t + \langle g^{(t)}, v - v^{(t)} \rangle$. Notice that since $\Psi(w^{(t)}, b_t, v; w', b')$ is a linear function of $v$, it is equal to this cut function, which therefore upper-bounds $\min_{w,b} \Psi(w, b, v; w', b')$.

One advantage of this cutting-plane formulation is that typical CutChooser implementations will choose $\epsilon_t$ to be large in the early iterations, and will only shrink it to be $\epsilon$ or smaller once we're close to convergence. We leave the details of the analysis to Appendices E and F—a summary can be found in Appendix G.

# 3   Related work

The problem of finding optimal trade-offs in the presence of multiple objectives has been studied generically in the field of multi-objective optimization [18]. Two common approaches are (i) linear scalarization [18, Section 3.1], and (ii) the method of $\epsilon$-constraints [18, Section 3.2]. Linear scalarization reduces to the common heuristic of reweighting groups of examples. The method of $\epsilon$-constraints puts hard bounds on the magnitudes of secondary objectives, like our dataset constraints. Notice that, in our formulation, the Lagrange multipliers $v$ play the role of the weights in the linear scalarization approach, with the difference being that, rather than being provided directly by the user, they are dynamically chosen to satisfy constraints. The user controls the problem through these constraint choices, which have concrete real-world meanings.

While the hinge loss is one of the most commonly-used convex upper bounds on the 0/1 loss [22], we use the ramp loss, trading off convexity for tightness. For our purposes, the main disadvantage of the hinge loss is that it is unbounded, and therefore cannot distinguish a single very bad example from say, 10 slightly bad ones, making it ill-suited for constraints on rates. In contrast, for the ramp loss the contribution of any single datum is bounded, no matter how far it is from the decision boundary.

The ramp loss has also been investigated in Collobert et al. [7] (without constraints). Gasso et al. [13] use the ramp loss both in the objective and constraints, but their algorithm only tackles the Neyman-Pearson problem. They compared their classifier to that of Davenport et al. [9], which differs in that it uses a hinge relaxation instead of the ramp loss, and found with the ramp loss they achieved similar or slightly better results with up to $10\times$ less computation (our approach does not enjoy this computational speedup).

Narasimhan et al. [19] considered optimizing the F-measure and other quantities that can be written as concave functions of the TP and TN rates. Their proposed stochastic dual solver adaptively linearizes concave functions of the rate functions (Equation 1). Joachims [16] indirectly optimizes upper-bounds on functions of $s_p(D^+)$, $s_p(D^-)$, $s_n(D^+)$, $s_n(D^-)$ using a hinge loss approximation.

Finally, for some simple problems (particularly when there is only one constraint), the goals in Section 1 can be coarsely handled by simple bias-shifting, i.e. first training an unconstrained classifier, and then attempting to adjust the decision threshold to satisfy the constraints as a second step.

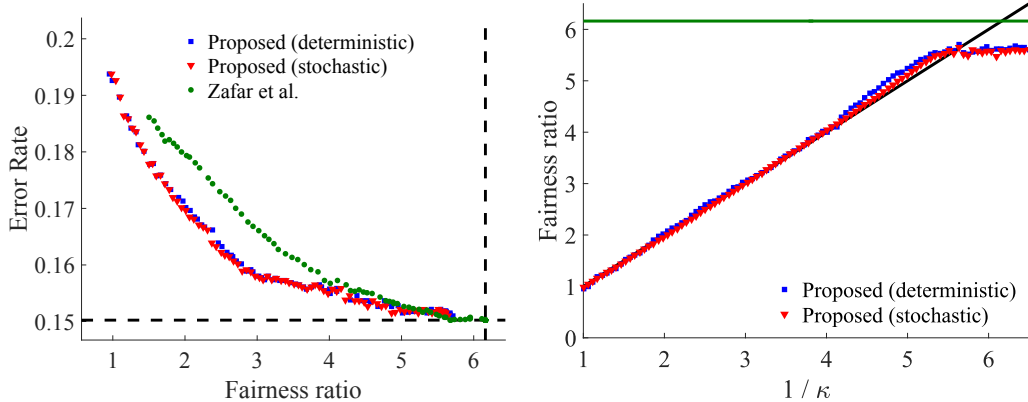

Figure 2: Blue dots: our proposal, with the classification functions' predictions being deterministically thresholded at zero. Red dots: same, but using the randomized classification rule described in Section 2. Green dots: Zafar et al. [27]. Green line: unconstrained SVM. **(Left)** Test set error plotted vs. observed test set fairness ratio $s_p\left(D^M\right)/s_p\left(D^F\right)$. **(Right)** The $1/\kappa$ hyper-parameter used to specify the desired fairness in the proposed method, and the observed fairness ratios of our classifiers on the test data. All points are averaged over 100 runs.

## 4 Experiments

We evaluate the performance of the proposed approach in two experiments, the first using a benchmark dataset for fairness, and the second on a real-world problem with churn and recall constraints.

### 4.1 Fairness

We compare training for fairness on the Adult dataset [2], the same dataset used by Zafar et al. [27]. The 32 561 training and 16 281 testing examples, derived from the 1994 Census, are 123-dimensional and sparse. Each feature contains categorical attributes such as race, gender, education levels and relationship status. A positive class label means that individual's income exceeds 50k. Let $D^M$ and $D^F$ denote the sets of male and female examples. The number of positive labels in $D^M$ is roughly six times that of $D^F$. The goal is to train a classifier that respects the fairness constraint $s_p\left(D^M\right) \leq s_p\left(D^F\right)/\kappa$ for a parameter $\kappa \in (0,1]$ (where $\kappa = 0.8$ corresponds to the 80% rule mentioned in Section 1).

Our publicly-available `Julia` implementation[3] for these experiments uses `LIBLINEAR` [11] with the default parameters (most notably $\lambda = 1/n \approx 3 \times 10^{-5}$) to implement the SVMOptimizer function, and does not include an unregularized bias $b$. The outer optimization over $v$ does not use the $m$-dimensional cutting plane algorithm of Algorithm 2, instead using a simpler one-dimensional variant (observe that these experiments involve only one constraint). The majorization-minimization procedure starts from the all-zeros vector ($w^{(0)}$ in Algorithm 1).

We compare to the method of Zafar et al. [27], which proposed handling fairness with the constraint:

$$\langle w, \bar{x} \rangle \leq c, \qquad \bar{x} = \left|D^M\right|^{-1}\sum\nolimits_{x \in D^M} x \; - \; \left|D^F\right|^{-1}\sum\nolimits_{x \in D^F} x. \tag{8}$$

An SVM subject to this constraint (see Appendix D for details), for a range of $c$ values, is our baseline.

Results in Figure 2 show the proposed method is much more accurate for any desired fairness, and achieves fairness ratios not reachable with the approach of Zafar et al. [27] for any choice of $c$. It is also easier to control: the values of $c$ in Zafar et al. [27] do not have a clear interpretation, whereas $\kappa$ is an effective proxy for the fairness ratio.

### 4.2 Churn

Our second set of experiments demonstrates meeting real-world requirements on a proprietary problem from Google: predicting whether a user interface element should be shown to a user, based

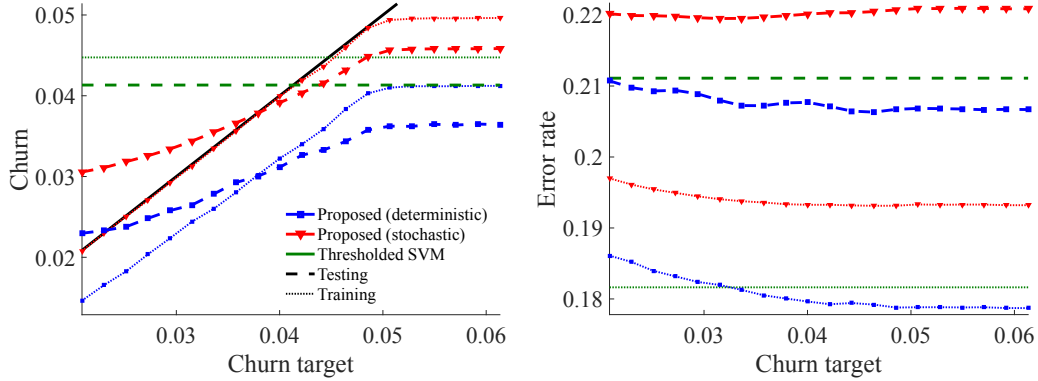

Figure 3: Blue: our proposal, with the classification functions' predictions being deterministically thresholded at zero. Red: same, but using the randomized classification rule described in Section 2. Green: unconstrained SVM trained on $D_1 \cup D_2$, then thresholded (by shifting the bias $b$) to satisfy the recall constraint on $D_2$. Dashed and dotted curves denote results on the testing and training datasets, respectively. **(Left)** Observed churn (vertical axis) vs. the churn target used during training (horizontal axis), on the unlabeled dataset $D_3$. **(Right)** Empirical error rates (vertical axis) vs. the churn target, on the union $D_1 \cup D_2$ of the two labeled datasets. All curves are averaged over 10 runs.

on a 31-dimensional vector of informative features, which is mapped to a roughly 30 000-dimensional feature vector via a fixed kernel function $\Phi$. We train classifiers that are linear with respect to $\Phi(x)$. We are given the currently-deployed model, and seek to train a classifier that (i) has high accuracy, (ii) has no worse recall than the deployed model, and (iii) has low churn w.r.t. the deployed model.

We are given three datasets, $D_1$, $D_2$ and $D_3$, consisting of 131 840, 53 877 and 68 892 examples, respectively. The datasets $D_1$ and $D_2$ are hand-labeled, while $D_3$ is unlabeled. In addition, $D_1$ was chosen via active sampling, while $D_2$ and $D_3$ are sampled *i.i.d.* from the underlying data distribution. For all three datasets, we split out 80% for training and reserved 20% for testing. We address the three goals in the proposed framework by simultaneously training the classifier to minimize the number of errors on $D_1$ plus the number of false positives on $D_2$, subject to the constraints that the recall on $D_2$ be at least as high as the deployed model's recall (we're essentially performing Neyman-Pearson classification on $D_2$), and that the churn w.r.t. the deployed model on $D_3$ be no larger than a given target parameter.

These experiments use a proprietary `C++` implementation of Algorithm 2, using the combined SDCA and cutting plane approach of Appendix F to implement the inner optimizations over $w$ and $b$, with the CutChooser helper functions being as described in Appendices E.1 and F.2.1. We performed 5 iterations of the majorization-minimization procedure of Algorithm 1.

Our baseline is an unconstrained SVM that is thresholded after training to achieve the desired recall, but makes no effort to minimize churn. We chose the regularization parameter $\lambda$ using a power-of-10 grid search, found that $10^{-7}$ was best for this baseline, and then used $\lambda = 10^{-7}$ for all experiments.

The plots in Figure 3 show the achieved churn and error rates on the training and testing sets for a range of churn constraint values (red and blue curves), compared to the baseline thresholded SVM (green lines). When using deterministic thresholding of the learned classifier (the blue curves, which significantly outperformed randomized classification–the red curves), the proposed method achieves lower churn and better accuracy for all targeted churn rates, while also meeting the recall constraint.

As expected, the empirical churn is extremely close to the targeted churn on the training set when using randomized classification (red dotted curve, left plot), but less so on the 20% held-out test set (red dashed curve). We hypothesize this disparity is due to overfitting, as the classifier has 30 000 parameters, and $D_3$ is rather small (please see Appendix C for a discussion of the generalization performance of our approach). However, except for the lowest targeted churn, the actual classifier churn (blue dashed curves) is substantially lower than the targeted churn. Compared to the thresholded SVM baseline, our approach significantly reduces churn without paying an accuracy cost.

## Footnotes

[1] Appendices may be found in the supplementary material

[2]"a9a" from `https://www.csie.ntu.edu.tw/~cjlin/libsvmtools/datasets/binary.html`

[3]`https://github.com/gabgoh/svmc.jl`

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
