[Supplementary Material]

Table 3: Key notation, listed in the order in which it was introduced.

| Symbol | Introduced | Description |
|---|---|---|
| $k$ | Section 2 | Number of datasets |
| $m$ | Section 2 | Number of dataset constraints |
| $D_i$ | Section 2 | $i$th dataset |
| $s_p, s_n$ | Section 2, Equation 1 | Positive and negative indicator-based rates |
| $\lambda$ | Section 2, Problem 1 | Regularization parameter |
| $\alpha_i^{(0)}, \beta_i^{(0)}$ | Section 2, Problem 1 | Coefficients defining the objective function |
| $\alpha_i^{(j)}, \beta_i^{(j)}$ | Section 2, Problem 1 | Coefficients defining the $j$th dataset constraint |
| $\gamma^{(j)}$ | Section 2, Problem 1 | Given upper bound of the $j$th dataset constraint |
| $\sigma$ | Section 2, Equation 2 | Ramp function: $\sigma(z) = \max\{0, \min\{1, \tfrac{1}{2} + z\}\}$ |
| $r_p, r_n$ | Section 2, Equation 3 | Positive and negative ramp-based rates |
| $\check{\sigma}_p, \check{\sigma}_n$ | Section 2.1, Equation 4 | Convex upper bounds on ramp functions |
| $\check{r}_p, \check{r}_n$ | Section 2.1, Equation 5 | Convex upper bounds on ramp-based rates |
| $\Psi$ | Section 2.2, Equation 7 | SVM objective (for minimizing over $w$ and $b$) |
| $F$ | Section 2.2, Equation 6 | Optimum of $\Psi$ (for maximizing over $v$) |
| $v$ | Section 2.2 | Lagrange multipliers associated with dataset constraints |
| $\mathcal{V}$ | Section 2.2, Algorithm 2 | Set of allowed $v$s |
| $v^{(s)}$ | Section 2.2, Algorithm 2 | Candidate solution at the $t$th iteration |
| $l_t, u_t$ | Section 2.2, Algorithm 2 | Lower and upper bounds on $F(v^{(t)})$ |
| $g^{(t)}$ | Section 2.2, Algorithm 2 | Gradient of the cutting plane inserted at the $t$th iteration |
| $h_t$ | Section 2.2, Algorithm 2 | Concave function upper-bounding $F(v)$ |
| $L_t, U_t$ | Section 2.2, Algorithm 2 | Lower and upper bounds on $\max_{v \in \mathcal{V}} F(v)$ |
| $V$ | Appendix C | Maximum allowed $v_j$: $\mathcal{V} \subseteq [0, V]^m$ |
| $\bar{s}_p, \bar{s}_n$ | Appendix C, Equation 10 | Expected positive and negative indicator-based rates |
| $\mu$ | Appendix E | Lebesgue measure |
| $S_\ell$ | Appendix E.2, Equation 11 | Superlevel set |
| $S_h$ | Appendix E.2, Equation 12 | Superlevel hypograph |
| $n$ | Appendix F.1 | Total size of datasets: $n = \sum_{i=1}^k |D_i|$ |
| $\check{\alpha}_i^{(0)}, \check{\beta}_i^{(0)}$ | Appendix F.1, Equation 13 | Coefficients defining the convex objective function |
| $\check{\alpha}_i^{(j)}, \check{\beta}_i^{(j)}$ | Appendix F.1, Equation 14 | Coefficients defining the $j$th convex dataset constraint |
| $\check{\gamma}^{(j)}$ | Appendix F.1, Equation 15 | Given upper bound of the $j$th convex dataset constraint |
| $\ell_{i,x}$ | Appendix F.1, Equation 16 | Loss of example $x$ in dataset $D_i$, in the SVM objective |
| $\check{\alpha}_i, \check{\beta}_i$ | Appendix F.1, Equation 17 | Coefficients defining the SVM objective function |
| $L$ | Appendix F.1, Equation 18 | Lipschitz constant of the $\ell_{i,x}$s |
| $\xi$ | Appendix F.1, Equation 19 | SVM dual variables |
| $\Psi^*$ | Appendix F.1, Equation 19 | SVM dual objective (for maximizing over $\xi$) |
| $b_s$ | Appendix F.2, Algorithm 3 | Candidate solution at the $t$th iteration |
| $l'_t, u'_t$ | Appendix F.2, Algorithm 3 | Lower and upper bounds on $\min_{w \in \mathbb{R}^d} \Psi(w, b_t, v; w', b')$ |
| $g'_t$ | Appendix F.2, Algorithm 3 | Derivative of the cutting plane inserted at the $t$th iteration |
| $h'_t$ | Appendix F.2, Algorithm 3 | Convex function lower-bounding $\min_{w \in \mathbb{R}^d} \Psi(w, b, v; w', b')$ |
| $L'_t, U'_t$ | Appendix F.2, Algorithm 3 | Lower and upper bounds on $\min_{b \in \mathcal{B}, w \in \mathbb{R}^d} \Psi(w, b, v; w', b')$ |

## A  Randomized classification

The use of the ramp loss in Problem 2 can be interpreted in two ways, which are exactly equivalent at training time, but lead to the use of different classification rules at evaluation time.

**Deterministic:** This is the obvious interpretation: we would like to optimize Problem 1, but cannot do so because the indicator-based rates $s_p$ and $s_n$ are discontinuous, so we approximate them with the ramp-based rates $r_n$ and $r_p$, and and hope that this approximation doesn't cost us too much, in terms of performance. The result is Problem 2. At evaluation time, on an example $x$, we make a positive prediction if $\langle w, x \rangle - b$ is nonnegative, and a negative prediction otherwise.

**Randomized:** In this interpretation (also used by Cotter et al. [8]), we reinterpret the ramp loss as the expected 0/1 loss suffered by a randomized classifier, with the result that the rates aren't being approximated *at all*—instead, we're using the indicator-based rates throughout, but randomizing the classifier and taking expectations to smooth out the discontinuities in the objective function. To be precise, at evaluation time, on an example $x$, we make a positive prediction with probability $\sigma(\langle w, x \rangle - b)$, and a negative prediction otherwise (with $\sigma$ being the ramp function of Equation 2).

Table 4: Some ratio metrics (Appendix B), which are metrics that can be written as ratios of linear combinations of rates. #Wins and #Losses are actually linear combination metrics, but are needed for the other definitions (as are Recall and #Changes from Table 2).

| Metric | Expression |
|---|---|
| Precision | $\#\mathrm{TP}/(\#\mathrm{TP}+\#\mathrm{FP})$ |
| $F_1$-score | $2\mathrm{Precision}\cdot\mathrm{Recall}/(\mathrm{Precision}+\mathrm{Recall})=2\#\mathrm{TP}/(2\#\mathrm{TP}+\#\mathrm{FN}+\#\mathrm{FP})$ |
| #Wins | $\left|D^{+-}\right|s_p\left(D^{+-}\right)+\left|D^{-+}\right|s_n\left(D^{-+}\right)$ |
| #Losses | $\left|D^{++}\right|s_n\left(D^{+-}\right)+\left|D^{--}\right|s_p\left(D^{-+}\right)$ |
| Win/loss Ratio | $\#\mathrm{Wins}/\#\mathrm{Losses}$ |
| Win/change Ratio | $\#\mathrm{Wins}/\#\mathrm{Changes}$ |

Taking expectations of the indicator-based rates $s_p$ and $s_n$ over the randomness of this classification rule yields the ramp-based rates $r_n$ and $r_p$, resulting, once again, in Problem 2.

This use of a randomized prediction isn't as unfamiliar as it may at first seem: in logistic regression, the classifier provides probability estimates at evaluation time (with $\sigma$ being a sigmoid instead of a ramp). Furthermore, at training time, the learned classifier is assumed to be randomized, so that the optimization problem can be interpreted as maximizing the data log-likelihood.

In the setting of this paper, the main advantages of the use of a randomized classification rule are that (i) we can say something about generalization performance (Appendix C), and (ii) because the rates are never being approximated, the dataset constraints will be satisfied *tightly* on the training dataset, in expectation (this is easily seen in the dotted red curve in the left plot of Figure 3). Despite these apparent advantages, deterministic classifiers seem to work better in practice.

## B  Ratio metrics

Problem 1 minimizes an objective function and imposes upper-bound constraints, all of which are written as linear combinations of positive and negative rates—we refer to such as "linear combination metrics". Some metrics of interest, however, cannot be written in this form. One important subclass are the so-called "ratio metrics", which are *ratios* of linear combinations of rates. Examples of ratio metrics are precision, $F_1$-score, win/loss ratio and win/change ratio (recall is a linear combination metric, since its denominator is a constant).

Ratio metrics may not be used directly in the objective function, but can be included in constraints by multiplying through by the denominator, then shifting the constraint coefficients to be non-negative. For example, the constraint that precision must be greater than $90\%$ can be expressed as follows:

$$\left|D^+\right|s_p\left(D^+\right)\geq 0.9\left(\left|D^+\right|s_p\left(D^+\right)+\left|D^-\right|s_p\left(D^-\right)\right)$$
$$0.1\left|D^+\right|s_p\left(D^+\right)-0.9\left|D^-\right|s_p\left(D^-\right)\geq 0$$
$$-0.1\left|D^+\right|s_p\left(D^+\right)+0.9\left|D^-\right|s_p\left(D^-\right)\leq 0$$
$$0.1\left|D^+\right|s_n\left(D^+\right)+0.9\left|D^-\right|s_p\left(D^-\right)\leq 0.1\left|D^+\right|,$$

where we used the fact that $s_p\left(D^+\right)+s_n\left(D^+\right)=1$ on the last line—this is an example of a fact that we noted in Section 2: since positive and negative rates must sum to one, it is possible to write any linear combination of rates as a positive linear combination, plus a constant.

Multiplying through by the denominator is fine for Problem 1, but a natural question is whether, by using a randomized classifier and optimizing Problem 2, we're doing the "right thing" in expectation. The answer is: not quite. Since the expectation of a ratio is not the ratio of expectations, e.g. a precision constraint in our original problem (Problem 1) becomes only a constraint on a precision-like quantity (the ratio of the expectations of the precision's numerator and denominator) in our relaxed problem.

## C  Generalization

In this appendix, we'll provide generalization bounds for an algorithm that is *nearly* identical to Algorithm 1. The two differences are that (i) we assume that the optimizer used on line 3 will prefer smaller biases $b$ to larger ones, i.e. that if Problem 3 has multiple equivalent minima, then the optimizer will return one for which $|b|$ is minimized, and (ii) that the Lagrange multipliers

are upper-bounded by a parameter $V \geq v_j$, i.e. that instead of optimizing Equation 6, line 3 of Algorithm 1 will optimize:

$$\max_{0 \preceq v \preceq V} \min_{w,b} \Psi(w, b, v; w', b'),\tag{9}$$

the difference being the upper bound on $v$. If $V$ is large enough that no $v_j$s are bound to a constraint, then this will have no effect on the solution. If, however, $V$ is too small, then the solution might not satisfy the dataset constraints. Notice that Algorithm 2 assumes that $v \in \mathcal{V}$, with $\mathcal{V}$ being compact—hence, for our proposed optimization procedure, the assumption is that $\mathcal{V} \subseteq [0, V]^m$.

With these assumptions in place, we're ready to move on to defining a function class that contains any solution that could be found by our algorithm, and bounding its Rademacher complexity.

**Lemma 1.** *Define $\mathcal{F}$ to be the set of all linear functions $f(x) = \langle w, x \rangle - b$ with $\|w\|_2 \leq XB/\lambda$ and $|b| \leq 1/2 + X^2B/\lambda$, where $X \geq \|x\|_2$ is a uniform upper bound on the magnitudes of all training examples, and:*

$$B = \sum_{i=1}^{k} \left( \alpha_i^{(0)} + \beta_i^{(0)} + V \sum_{j=1}^{m} \left( \alpha_i^{(j)} + \beta_i^{(j)} \right) \right).$$

*Then $\mathcal{F}$ will contain all $|b|$-minimizing optimal solutions of Equation 9 for any $(w', b')$ and any training dataset.*

*Proof.* Let $f(w, b) + (\lambda/2) \|w\|_2^2$ be the the the objective function of Problem 3, and $g_j(w, b) \leq \gamma^{(j)}$ the $j$th constraint. Then it follows that:

$$\|\nabla_w f(w, b)\|_2 \leq X \sum_{i=1}^{k} \left( \alpha_i^{(0)} + \beta_i^{(0)} \right)$$

$$\|\nabla_w g_j(w, b)\|_2 \leq XV \sum_{i=1}^{k} \left( \alpha_i^{(j)} + \beta_i^{(j)} \right).$$

Differentiating the definition of $\Psi$ (Equation 7) and setting the result equal to zero shows that any optimal $w$ must satisfy (this is the stationarity KKT condition):

$$\lambda w = -\nabla_w f(w, b) - \sum_{j=1}^{m} v_j \nabla_w g_j(w, b),$$

implying by the triangle inequality that $\|w\|_2 \leq XB/\lambda$, where $B$ is as defined in the theorem statement.

Now let's turn our attention to $b$. The above bound implies that, if $w$ is optimal, then $|\langle w, x \rangle| \leq X^2B/\lambda$, from which it follows that the hinge functions $\max\{0, 1/2 + (\langle w, x \rangle - b)\}$ and $\max\{0, 1/2 - (\langle w, x \rangle - b)\}$ will be nondecreasing in $|b|$ as long as $|b| > 1/2 + X^2B/\lambda$. Problem 3 seeks to minimize a positive linear combination of such hinge functions subject to upper-bound constraints on positive linear combinations of such hinge functions, so our assumption that the optimizer used on line 3 of Algorithm 1 will always choose the smallest optimal $b$ gives that $|b| \leq 1/2 + X^2B/\lambda$. $\square$

**Lemma 2.** *The function class $\mathcal{F}$ of Lemma 1 has Rademacher complexity [2]:*

$$\mathcal{R}_n(\mathcal{F}) \leq \frac{1}{2\sqrt{n}} + \frac{2X^2}{\lambda\sqrt{n}} \sum_{i=1}^{k} \left( \alpha_i^{(0)} + \beta_i^{(0)} + V \sum_{j=1}^{m} \left( \alpha_i^{(j)} + \beta_i^{(j)} \right) \right),$$

*where $X \geq \|x\|_2$, as in Lemma 1, is a uniform upper bound on the magnitudes of all training examples.*

*Proof.* The Rademacher complexity of $\mathcal{F}$ is:

$$\mathcal{R}_n(\mathcal{F}) = \mathbb{E} \left[ \sup_{f \in \mathcal{F}} \frac{1}{n} \sum_{i=1}^{n} \epsilon_i f(x_i) \right]$$

$$= \mathbb{E} \left[ \sup_{w: \|w\|_2 \leq \frac{XB}{\lambda}} \frac{1}{n} \sum_{i=1}^{n} \epsilon_i \langle w, x \rangle \right] + \mathbb{E} \left[ \sup_{b: |b| \leq \frac{1}{2} + \frac{X^2B}{\lambda}} \frac{1}{n} \sum_{i=1}^{n} \epsilon_i b \right],$$

where the expectations are taken over the *i.i.d.* Rademacher random variables $\epsilon_1, \ldots, \epsilon_n$ and the *i.i.d.* training sample $x_1, \ldots, x_n$, and $B$ is as in Lemma 1. Applying the Khintchine inequality and substituting the definition of $B$ yields the claimed bound. □

We can now apply the results of Bartlett and Mendelson [2] to prove bounds on the generalization error. To this end, we assume that each of our training datasets $D_i$ is drawn *i.i.d.* from some underlying unknown distribution $\mathcal{D}_i$. We will bound the expected positive and negative prediction rates w.r.t. these distributions:

$$\bar{s}_p(\mathcal{D}; f) = \mathbb{E}_{x \sim \mathcal{D}}[f(x)] \qquad \bar{s}_n(\mathcal{D}; f) = \bar{s}_p(\mathcal{D}; 1 - f), \tag{10}$$

where $f : \mathbb{R}^d \to \{0, 1\}$ is a binary classification function.

**Theorem 1.** *For a given $(w, b)$ pair, define $f_{w,b}(x)$ such that it predicts $1$ with probability $\sigma(\langle w, x \rangle - b)$, and $0$ otherwise ($\sigma$ is as in Equation 2, so this is the randomized classifier of Appendix A).*

*Suppose that the $k$ training datasets $D_i$ have sizes $n_i = |D_i|$, and that $D_i$ is drawn i.i.d. from $\mathcal{D}_i$ for all $i \in \{1, \ldots, k\}$. Then, with probability $1 - \delta$ over the training samples, uniformly over all $(w, b)$ pairs that are optimal solutions of Equation 9 for some $(w', b')$ under the assumptions listed at the start of this appendix, the expected rates will satisfy:*

$$\bar{s}_p(\mathcal{D}_i; f_{w,b}) \leq r_p(D_i; w, b) + E/\sqrt{n_i}$$
$$\bar{s}_n(\mathcal{D}_i; f_{w,b}) \leq r_n(D_i; w, b) + E/\sqrt{n_i},$$

*the above holding for all $i \in \{1, \ldots, k\}$, where:*

$$E = 1 + \frac{4X^2}{\lambda} \sum_{i=1}^{k} \left( \alpha_i^{(0)} + \beta_i^{(0)} + V \sum_{j=1}^{m} \left( \alpha_i^{(j)} + \beta_i^{(j)} \right) \right) + \sqrt{8 \ln \left( \frac{4k}{\delta} \right)},$$

*with $X \geq \|x\|_2$, as in Lemmas 1 and 2, being a uniform upper bound on the magnitudes of all training examples $x \sim \mathcal{D}_i$ for all $i \in \{1, \ldots, k\}$.*

*Proof.* Observe that the ramp rates $r_p$ and $r_n$ are 1-Lipschitz. Applying Theorems 8 and 12 (part 4) of Bartlett and Mendelson [2] gives that each of the following inequalities hold with probability $1 - \delta/2k$, for all $i \in \{1, \ldots, k\}$:

$$\mathbb{E}_{x \sim \mathcal{D}_i}[r_p(\{x\}; w, b)] \leq r_p(D_i; w, b) + 2\mathcal{R}_{n_i}(\mathcal{F}) + \sqrt{\frac{8}{n_i} \ln \left( \frac{4k}{\delta} \right)}$$

$$\mathbb{E}_{x \sim \mathcal{D}_i}[r_n(\{x\}; w, b)] \leq r_n(D_i; w, b) + 2\mathcal{R}_{n_i}(\mathcal{F}) + \sqrt{\frac{8}{n_i} \ln \left( \frac{4k}{\delta} \right)},$$

where $\mathcal{R}_n(\mathcal{F})$ is as in Lemma 2. The union bound implies that all $2k$ inequalities hold simultaneously with probability $1 - \delta$. The LHSs above are the expected ramp-based rates of a deterministic classifier, but as was explained in Appendix A, these are identical to the expected indicator-based rates of a randomized classifier, which is what is claimed. □

An immediate consequence of this result is that (with probability $1 - \delta$) if $(w, b)$ suffers the training loss:

$$\hat{\mathcal{L}} = \sum_{i=1}^{k} \left( \alpha_i^{(0)} r_p(D_i; w, b) + \beta_i^{(0)} r_n(D_i; w, b) \right),$$

then the expected loss on previously-unseen data (drawn *i.i.d.* from the same distributions) will be upper-bounded by:

$$\hat{\mathcal{L}} + E \sum_{i=1}^{k} \frac{\alpha_i^{(0)} + \beta_i^{(0)}}{\sqrt{n_i}}.$$

Likewise, if $(w, b)$ satisfies the constraint:

$$\sum_{i=1}^{k} \left( \alpha_i^{(j)} r_p(D_i; w, b) + \beta_i^{(j)} r_n(D_i; w, b) \right) \leq \gamma^{(j)},$$

then the corresponding rate constraint on previously-unseen data will be violated by no more than:

$$E \sum_{i=1}^{k} \frac{\alpha_i^{(j)} + \beta_i^{(j)}}{\sqrt{n_i}}$$

in expectation, where, here and above, $E$ is as in Theorem 1.

## D   Fairness constraints of Zafar et al. [27]

The constraints of Zafar et al. [27] can be interpreted as a relaxation of the constraint $-c \leq s_p(D^A; w) - s_p(D^B; w) \leq c$ under the linear approximation

$$s_p(D; w, b) \approx \frac{1}{|D|} \sum_{x \in D} (\langle w, x \rangle - b),$$

giving:

$$s_p(D^A; w, b) - s_p(D^B; w, b) \approx \frac{1}{|D^A|} \sum_{x \in D^A} (\langle w, x \rangle - b) - \frac{1}{|D^B|} \sum_{x \in D^B} (\langle w, x \rangle - b) = \langle w, \bar{x} \rangle,$$

where $\bar{x}$ is defined as in Equation 8. We can therefore implement the approach of Zafar et al. [27] within our framework by adding the constraints:

$$\langle w, \bar{x} \rangle \leq c \iff \max\{0, 1 - \langle w, \bar{x} \rangle\} \leq c + 1$$
$$c \leq \langle w, \bar{x} \rangle \iff \max\{0, 1 + \langle w, \bar{x} \rangle\} \leq c + 1,$$

and solving the hinge constrained optimization problem described in Problem 3. Going further, we could implement these constraints as egregious examples using the constraint:

$$\langle w, \bar{x} \rangle \leq c \iff \left\langle w, \frac{1}{4c} \bar{x} \right\rangle \leq \frac{1}{4} \iff \frac{1}{2} + \left\langle w, \frac{1}{4c} \bar{x} \right\rangle \leq \frac{3}{4}$$

$$\iff \min\left\{ \max\left\{ \frac{1}{2} + \left\langle w, \frac{1}{4c} \bar{x} \right\rangle, 0 \right\}, 1 \right\} \leq \frac{3}{4} \iff r_p(\bar{x}) \leq \frac{3}{4},$$

permitting us to perform an analogue of their approximations in ramp form.

## E   Cutting plane algorithm

We'll now discuss some variants of Algorithm 2. We assume that $F(v)$ is the function that we wish to maximize for $v \in \mathcal{V}$, where:

1. $\mathcal{V} \subseteq \mathbb{R}^m$ is compact and convex.
2. $F : \mathcal{V} \to \mathbb{R}$ is concave.
3. $F$ has a (not necessarily unique) maximizer $v^* = \operatorname{argmax}_{v \in \mathcal{V}} F(v)$.

For the purposes of Algorithm 2, we would take $F$ to be as in Equation 6, but the same approach can be applied more generally.

### E.1   Maximization-based

We're primarily interested in proving convergence rates, and will do so in Appendix E.2. With that said, there is one easy-to-implement variant of Algorithm 2 for which we have not proved a convergence rate, but that we use in some of our experiments due to its simplicity:

**Definition 1.** *(Maximization-based Algorithm 2) CutChooser chooses* $v^{(t)} = \operatorname{argmax}_{v \in \mathcal{V}} h_t(v)$ *and* $\epsilon_t = (U_t - L_t)/2$.

Observe that this $v^{(t)}$ can be found at the same time as $U_t$ is computed, since both result from optimization of the same linear program. However, despite the ease of implementing this variant, we have not proved any convergence rates about it.

### E.2   Center of mass-based

We'll now discuss a variant of Algorithm 2 that chooses $v^{(t)}$ and $\epsilon_t$ based on the center of mass of the "superlevel hypograph" determined by $h_t$ and $L_t$, which we define as the intersection of

the hypograph of $h_t$ (the set of $m+1$-dimensional points $(v, z)$ for which $z \le h_t(z)$) and the half-space containing all points $(v, z)$ for which $z \ge L_t$. Notice that, in the context of Algorithm 2, the superlevel hypograph defined by $h_t$ and $L_t$ corresponds to the set of pairs of candidate maximizers and their possible function values at the $t$th iteration. Because this variant is based on finding a cut center in the $m+1$-dimensional hypograph, rather than an $m$-dimensional level set (which is arguably more typical), this is an instance of what Boyd and Vandenberghe [5] call an "epigraph cutting plane method".

Throughout this section, we will take $\mu$ to be the Lebesgue measure (either 1-dimensional, $m$-dimensional, or $m+1$-dimensional, depending on context). We also must define some notation for dealing with superlevel sets and hypographs. For a concave $f : \mathcal{V} \to \mathbb{R}$ and $y \in \mathbb{R}$, define:

$$S_\ell(f, y) = \{v \in \mathcal{V} \mid f(v) \ge y\} \tag{11}$$

as the superlevel set of $f$ at $y$. Further define:

$$S_h(f, y) = \{(v, z) \in \mathcal{V} \times \mathbb{R} \mid f(v) \ge z \ge y\} \tag{12}$$

as the superlevel hypograph of $f$ above $y$. With these definitions in place, we're ready to explicitly state the center of mass-based rule for the CutChooser function on line 8 of Algorithm 2:

**Definition 2.** *(Center of mass-based Algorithm 2) CutChooser takes $(v^{(t)}, z_t)$ to be the center of mass of $S_h(h_t, L_t)$, and chooses $\epsilon_t = (z_t - L_t)/2$.*

Finding the center of mass of a polytope is a difficult problem in general [20, 21], so our convergence results for this version of CutChooser are mostly of theoretical interest. With that said, for one dimensional problems (the setting of Appendix F.2) it may be implemented efficiently.

Our final bit of "setup" before getting to our results is to state two classic theorems, plus a corollary, which will be needed for our proofs. The first enables us to interpolate the areas of superlevel sets:

**Theorem 2.** *Suppose that the superlevel sets of a concave $f : \mathcal{V} \to \mathbb{R}$ at $y_1$ and $y_2$ are nonempty, and take $\gamma \in [0, 1]$. Then:*

$$\left(\mu\left(S_\ell\left(f, \gamma y_1 + (1-\gamma) y_2\right)\right)\right)^{1/m} \ge \gamma \left(\mu\left(S_\ell\left(f, y_1\right)\right)\right)^{1/m} + (1-\gamma) \left(\mu\left(S_\ell\left(f, y_2\right)\right)\right)^{1/m}.$$

*Proof.* This is the Brunn-Minkowski inequality [e.g. 1]. □

This theorem has the immediate useful corollary:

**Corollary 1.** *Suppose that $f : \mathcal{V} \to \mathbb{R}$ is concave with a maximizer $v^* \in \mathcal{V}$, and that $\delta \ge 0$. Then:*

$$\left(\frac{\delta}{m+1}\right) \mu\left(S_\ell\left(f, f(v^*) - \delta\right)\right) \le \mu\left(S_h\left(f, f(v^*) + \delta\right)\right) \le \delta \mu\left(S_\ell\left(f, f(v^*) - \delta\right)\right).$$

*Proof.* By Theorem 2 (lower-bounding the second term on the RHS by zero), for $0 \le z \le \delta$:

$$\mu\left(S_\ell\left(f, f(v^*) - z\right)\right) \ge \left(\frac{z}{\delta}\right)^m \mu\left(S_\ell\left(f, f(v^*) - \delta\right)\right),$$

from which integrating $\mu\left(S_h\left(f, f(v^*) - \delta\right)\right) = \int_0^\delta \mu\left(S_\ell\left(f, f(v^*) - z\right)\right) m\mu(z)$ yields the claimed lower bound. The upper bound follows immediately from the fact that the superlevel sets shrink as $z$ increases (i.e. $\mu\left(S_\ell\left(f, z'\right)\right) \le \mu\left(S_\ell\left(f, z\right)\right)$ for $z' \ge z$). □

The second classic result enables us to bound how much "progress" is made by a cut based on the center of mass of a superlevel hypograph:

**Theorem 3.** *Suppose that $S \subseteq \mathbb{R}^m$ is a convex set. If we let $z \in S$ be the center of mass of $S$, then for any half-space $H \ni z$:*

$$\frac{\mu(S \cap H)}{\mu(S)} \ge \left(\frac{m}{m+1}\right)^m \ge \frac{1}{e}.$$

*Proof.* This is Theorem 2 of Grünbaum [14]. □

With the preliminaries out of the way, we're ready to move on to our first result: bounding the volumes of the superlevel hypographs of our $h_t$s, assuming that we base our cuts on the centers of mass of the superlevel hypographs:

**Lemma 3.** *In the context of Algorithm 2, suppose that we choose $v^{(t)}$ and $\epsilon_t$ as in Definition 2. Then:*

$$\mu\left(S_h\left(h_{t+1}, L_{t+1}\right)\right) \leq \left(1 - \frac{1}{2e}\right)\mu\left(S_h\left(h_t, L_t\right)\right),$$

*from which it follows that:*

$$\mu\left(S_h\left(h_t, L_t\right)\right) \leq \left(1 - \frac{1}{2e}\right)^{t-1}\left(u_0 - l_0\right)\mu\left(\mathcal{V}\right),$$

*for all $t$.*

*Proof.* We'll consider two cases: $u_t \leq z_t$ and $u_t > z_t$, corresponding to making a "deep" or "shallow" cut, respectively.

*Deep cut case:* If $u_t \leq z_t$, then the hyperplane $u_t + \left\langle g^{(t)}, v - v^{(t)}\right\rangle$ passes below the center of mass of $S_h(h_t, L_t)$, implying by Theorem 3 that:

$$\mu\left(S_h\left(h_{t+1}, L_{t+1}\right)\right) \leq \mu\left(S_h\left(h_{t+1}, L_t\right)\right) \leq \left(1 - \frac{1}{e}\right)\mu\left(S_h\left(h_t, L_t\right)\right).$$

*Shallow cut case:* Now suppose that $u_t > z_t$. Applying Theorem 3 to the level cut $\{(v, z) \mid z \leq z_t\}$ at $z_t$:

$$\frac{1}{e}\mu\left(S_h\left(h_t, L_t\right)\right) \leq \int_{L_t}^{z_t} \mu\left(\{v \in \mathcal{V} \mid h_t\left(v\right) \geq z\}\right) d\mu(z)$$

$$\leq \int_{L_t}^{(z_t+L_t)/2} \mu\left(\{v \in \mathcal{V} \mid h_t\left(v\right) \geq z\}\right) d\mu(z)$$

$$+ \int_{(z_t+L_t)/2}^{z_t} \mu\left(\{v \in \mathcal{V} \mid h_t\left(v\right) \geq z\}\right) d\mu(z).$$

Since $h_t$ is concave, its superlevel sets shrink for larger $z$, so the first integral on the RHS above is larger than the second, implying that:

$$\frac{1}{2e}\mu\left(S_h\left(h_t, L_t\right)\right) \leq \int_{L_t}^{(z_t+L_t)/2} \mu\left(\{v \in \mathcal{V} \mid h_t\left(v\right) \geq z\}\right) d\mu(z).$$

The fact that $\epsilon_t = (z_t - L_t)/2$ implies that $l_t > (z_t + L_t)/2$, so $L_{t+1} > (z_t + L_t)/2$, and:

$$\frac{1}{2e}\mu\left(S_h\left(h_t, L_t\right)\right) \leq \int_{L_t}^{L_{t+1}} \mu\left(\{v \in \mathcal{V} \mid h_t\left(v\right) \geq z\}\right) d\mu(z),$$

showing that we will cut off at least a $1/2e$-proportion of the total volume, completing the proof of the first claim.

The second claim follows immediately by iterating the first, and observing that $\mu\left(S_h\left(h_1, L_1\right)\right) = \left(u_0 - l_0\right)\mu\left(\mathcal{V}\right)$. □

The above result shows that the volumes of the superlevel hypographs of the $h_t$s shrink at an exponential rate. However, our actual stopping condition (line 5 of Algorithm 2) depends not on the volume, but rather the "height" $U_t - L_t$, so we would prefer a bound on this height, rather than the volume. We find such a bound in the (proof of the) following lemma, which establishes how many iterations must elapse before the stopping condition is satisfied:

**Lemma 4.** *In the context of Algorithm 2, suppose that we choose $v^{(t)}$ and $\epsilon_t$ as in Definition 2. Then there is a iteration count $T_\epsilon$ satisfying:*

$$T_\epsilon = O\left(m \ln\left(\frac{u_0 - l_0}{\epsilon}\right) + \ln\left(\frac{\mu\left(\mathcal{V}\right)}{\mu\left(S_\ell\left(F, l_0\right)\right)}\right)\right),$$

*such that, if $t \geq T_\epsilon$, then $U_t - L_t \leq \epsilon$. Hence, Algorithm 2 will terminate after $T_\epsilon$ iterations.*

*Proof.* By Corollary 1:

$$\mu\left(S_h\left(h_t, L_t\right)\right) \geq \left(\frac{U_t - L_t}{m+1}\right)\mu\left(S_\ell\left(h_t, L_t\right)\right).$$

If $L_t \leq F\left(v^*\right) - \epsilon$, then $\mu\left(S_\ell\left(h_t, L_t\right)\right) \geq \mu\left(S_\ell\left(h_t, F\left(v^*\right) - \epsilon\right)\right)$ because $h_t$ is concave. If $L_t > F\left(v^*\right) - \epsilon$, then by Theorem 2:

$$\mu\left(S_\ell\left(h_t, L_t\right)\right) \geq \left(\frac{U_t - L_t}{U_t - F\left(v^*\right) + \epsilon}\right)^m \mu\left(S_\ell\left(h_t, F\left(v^*\right) - \epsilon\right)\right).$$

In either case, $L_t \leq F\left(v^*\right)$ by definition, and we'll assume that $U_t - L_t > \epsilon$ (this will lead to a contradiction), so:

$$\mu\left(S_h\left(h_t, L_t\right)\right) \geq 2^{-m}\left(\frac{U_t - L_t}{m+1}\right)\mu\left(S_\ell\left(h_t, F\left(v^*\right) - \epsilon\right)\right).$$

Applying Lemma 3 yields that:

$$\left(1 - \frac{1}{2e}\right)^{t-1}\left(u_0 - l_0\right)\mu\left(\mathcal{V}\right) \geq 2^{-m}\left(\frac{U_t - L_t}{m+1}\right)\mu\left(S_\ell\left(h_t, F\left(v^*\right) - \epsilon\right)\right).$$

Next observe that, by Theorem 2:

$$\mu\left(S_\ell\left(h_t, F\left(v^*\right) - \epsilon\right)\right) \geq \left(\frac{U_t - F\left(v^*\right) + \epsilon}{U_t - l_0}\right)^m \mu\left(S_\ell\left(h_t, l_0\right)\right) \geq \left(\frac{\epsilon}{u_0 - l_0}\right)^m \mu\left(S_\ell\left(F, l_0\right)\right).$$

Combining the previous two equations gives:

$$U_t - L_t \leq \left(1 - \frac{1}{2e}\right)^{t-1}\left(m+1\right)\left(\frac{2}{\epsilon}\right)^m \left(u_0 - l_0\right)^{m+1}\left(\frac{\mu\left(\mathcal{V}\right)}{\mu\left(S_\ell\left(F, l_0\right)\right)}\right).$$

Simplifying this inequality yields that, if we have performed the claimed number of iterations, then $U_t - L_t \leq \epsilon$ (this contradicts our earlier assumption that $U_t - L_t > \epsilon$, so this is technically a proof by contradiction). □

The second term in the bound on $T_\epsilon$ measures how closely $\mathcal{V}$ matches with the set of all points $z$ on which $F\left(z\right)$ exceeds our initial lower bound $l_0$. Observe that if $l_0 \leq F\left(v\right)$ for all $v \in \mathcal{V}$, then $\mu\left(S_\ell\left(F, l_0\right)\right) = \mu\left(\mathcal{V}\right)$, and this term will vanish.

Bounding the number of cutting-plane iterations that will be performed is not enough to establish how quickly our procedure will converge, since we rely on performing an inner SVM optimizations with target suboptimality $\epsilon_t$, and the runtime of these inner optimizations naturally will depend on the magnitudes of the $\epsilon_t$s, which are bounded in our final lemma:

**Lemma 5.** *In the context of Algorithm 2, suppose that we choose $v^{(t)}$ and $\epsilon_t$ as in Definition 2. Then:*

$$\epsilon_t \geq \frac{U_t - L_t}{2e\left(m+1\right)},$$

*and in particular, for all $t$ (before termination):*

$$\epsilon_t \geq \frac{\epsilon}{2e\left(m+1\right)},$$

*since we terminate as soon as $U_t - L_t \leq \epsilon$.*

*Proof.* Because $h_t$ is concave:

$$\mu\left(S_h\left(h_t, L_t\right)\right) - \mu\left(S_h\left(h_t, z_t\right)\right) \leq \left(z_t - L_t\right)\mu\left(S_\ell\left(h_t, L_t\right)\right),$$

where $z_t$ is as in Lemma 3. By Corollary 1, $\mu\left(S_\ell\left(h_t, L_t\right)\right) \leq \frac{m+1}{U_t - L_t}\mu\left(S_h\left(h_t, L_t\right)\right)$, which combined with the above inequality gives that:

$$\frac{\mu\left(S_h\left(h_t, L_t\right)\right) - \mu\left(S_h\left(h_t, z_t\right)\right)}{\mu\left(S_h\left(h_t, L_t\right)\right)} \leq \frac{z_t - L_t}{U_t - L_t}\left(m+1\right).$$

By Theorem 3, the LHS is at least $1/e$, and $z_t - L_t = 2\epsilon_t$, giving the claimed result. □

# F SVM optimization

We'll now move onto a discussion of how we propose implementing the SVMOptimizer of Algorithm 2. The easier-to-analyze approach, based on an inner SDCA optimization over $w$ [24] and an outer cutting plane optimization over $b$ (Algorithm 3), will be described in Appendices F.1 and F.2. The easier-to-implement version, which simply calls an off-the-shelf SVM solver, will be described in Appendix F.3.

## F.1 SDCA $w$-optimization

To simplify the presentation, we're going to begin by reformulating Equation 7 in such a way that all of the datasets are "mashed together", with the coefficients being defined on a per-example basis, rather than per-dataset. To this end, for fixed $w'$ and $b'$, we define, for every $i \in \{1, \ldots, k\}$ and every $x \in D_i$:

$$\check{\alpha}_{i,x}^{(0)} = \begin{cases} \alpha_i^{(0)} & \text{if } \langle w', x \rangle - b' \leq 1/2 \\ 0 & \text{otherwise} \end{cases} \tag{13}$$

$$\check{\beta}_{i,x}^{(0)} = \begin{cases} \beta_i^{(0)} & \text{if } \langle w', x \rangle - b' \geq -1/2 \\ 0 & \text{otherwise} \end{cases}.$$

This takes care of the loss coefficients. For the constraint coefficients, define:

$$\check{\alpha}_{i,x}^{(j)} = \begin{cases} \alpha_i^{(j)} & \text{if } \langle w', x \rangle - b' \leq 1/2 \\ 0 & \text{otherwise} \end{cases} \tag{14}$$

$$\check{\beta}_{i,x}^{(j)} = \begin{cases} \beta_i^{(j)} & \text{if } \langle w', x \rangle - b' \geq -1/2 \\ 0 & \text{otherwise} \end{cases}.$$

and finally, we need to handle the constraint upper bounds:

$$\check{\gamma}^{(j)} = \gamma^{(j)} - \sum_{i=1}^{k} \frac{1}{|D_i|} \left( \alpha_i^{(j)} \left| \{x \in D_i \mid \langle w', x \rangle - b' > 1/2\} \right| \right. \tag{15}$$

$$\left. + \beta_i^{(j)} \left| \{x \in D_i \mid \langle w', x \rangle - b' < -1/2\} \right| \right).$$

Observe that the $\check{\alpha}_{i,x}^{(0)}$s, $\check{\beta}_{i,x}^{(0)}$s, $\check{\alpha}_{i,x}^{(j)}$s, $\check{\beta}_{i,x}^{(j)}$s, and $\check{\gamma}^{(j)}$s all have implicit dependencies on $w'$ and $b'$. In terms of these definitions, the $\Psi$ defined in Equation 7 can be written as:

$$\Psi(w, b, v; w', b') = \sum_{i=1}^{k} \frac{1}{|D_i|} \sum_{x \in D_i} \left( \left( \check{\alpha}_{i,x}^{(0)} + \sum_{j=1}^{m} v_j \check{\alpha}_{i,x}^{(j)} \right) \max \left\{ 0, \frac{1}{2} + (\langle w, x \rangle - b) \right\} \right.$$

$$\left. + \left( \check{\beta}_{i,x}^{(0)} + \sum_{j=1}^{m} v_j \check{\beta}_{i,x}^{(j)} \right) \max \left\{ 0, \frac{1}{2} - (\langle w, x \rangle - b) \right\} \right)$$

$$+ \frac{\lambda}{2} \|w\|_2^2 - \sum_{j=1}^{m} v_j \check{\gamma}^{(j)}.$$

This formulation makes it clear that minimizing $\Psi$ as a function of $w$ and $b$ is equivalent to optimizing an SVM, since $\Psi$ is just a positive linear combination of hinge losses, plus a $\ell^2$ regularizer, plus a term that does not depend on $w$ or $b$. Since $\Psi$ can have both "positive" and "negative" hinge losses associated with the same example, however, it's slightly simpler to combine both hinge losses together into a single piecewise linear per-example loss, rather than decomposing it into two separate hinges:

$$\ell_{i,x}(z) = \check{\alpha}_{i,x} \max \left\{ 0, \frac{1}{2} + z \right\} + \check{\beta}_{i,x} \max \left\{ 0, \frac{1}{2} - z \right\}, \tag{16}$$

where:

$$\check{\alpha}_{i,x} = \frac{n}{|D_i|} \left( \check{\alpha}_{i,x}^{(0)} + \sum_{j=1}^{m} v_j \check{\alpha}_{i,x}^{(j)} \right) \quad \text{and} \quad \check{\beta}_{i,x} = \frac{n}{|D_i|} \left( \check{\beta}_{i,x}^{(0)} + \sum_{j=1}^{m} v_j \check{\beta}_{i,x}^{(j)} \right). \tag{17}$$

Here, $n = \sum_{i=1}^{k} |D_i|$ is the total number of examples across all of the datasets—we introduced the $n$ factor here so that $\Psi$ will be written in terms of the *average* loss (rather than the *total* loss). Although it is not represented explicitly in our notation, it should be emphasized that $\ell_{i,x}$ implicitly depends on $v$, $w'$ and $b'$.

As the sum of two hinges, the $\ell_{i,x}$s are Lipschitz continuous in $z$, with the Lipschitz constant being:

$$L = \max_{i \in \{1,\ldots,k\}} \frac{n}{|D_i|} \left( \left( \alpha_i^{(0)} + \beta_i^{(0)} \right) + \sum_{j=1}^{m} v_j \left( \alpha_i^{(j)} + \beta_i^{(j)} \right) \right). \qquad (18)$$

Notice that, if the datasets are comparable in size, then $n/|D_i|$ will be on the order of $k$, so $L$ will typically not be as large as the $n$-dependence of its definition would appear to imply.

We may now write $\Psi$ in terms of the loss functions $\ell_{i,x}$:

$$\Psi(w, b, v; w', b') = \frac{1}{n} \sum_{i=1}^{k} \sum_{x \in D_i} \ell_{i,x} (\langle w, x \rangle - b) + \frac{\lambda}{2} \|w\|_2^2 - \sum_{j=1}^{m} v_j \tilde{\gamma}^{(j)}.$$

This is the form considered by Shalev-Shwartz and Zhang [24], so we may apply SDCA:

**Theorem 4.** *If we use SDCA [24] to optimize Equation 19 for fixed $b$ and $v$, then we will find a suboptimal solution with duality gap $\epsilon''$ after performing at most:*

$$T_{\epsilon''} = O \left( \max \left\{ 0, n \ln \left( \frac{\lambda n}{L^2 X^2} \right) \right\} + n + \frac{L^2 X^2}{\lambda \epsilon''} \right)$$

*iterations, where $X = \max_{i \in \{1,\ldots,k\}} \max_{x \in D_i} \|x\|_2$ is a uniform upper bound on the norms of the training examples.*

*Proof.* This is Theorem 2 of Shalev-Shwartz and Zhang [24]. $\qquad\square$

SDCA works by, rather than directly minimizing $\Psi$ over $w$, instead maximizing the following over the dual variables $\xi$:

$$\Psi^* (\xi, b, v; w', b') = \qquad (19)$$

$$- \frac{1}{n} \sum_{i=1}^{k} \sum_{x \in D_i} \ell_{i,x}^* (\xi_{i,x}) - \frac{1}{2\lambda} \left\| \frac{1}{n} \sum_{i=1}^{k} \sum_{x \in D_i} \xi_{i,x} x \right\|_2^2 - \frac{1}{n} \sum_{i=1}^{k} \sum_{x \in D_i} \xi_{i,x} b - \sum_{j=1}^{m} v_j \tilde{\gamma}^{(j)},$$

using stochastic coordinate ascent, where:

$$w = -\frac{1}{\lambda n} \sum_{i=1}^{k} \sum_{x \in D_i} \xi_{i,x} x$$

is the primal solution $w$ corresponding to a given set of dual variables $\xi$, and:

$$\ell_{i,x}^* (\xi_{i,x}) = \frac{1}{2} \left| \xi_{i,x} - \check{\alpha}_{i,x} + \check{\beta}_{i,x} \right| - \frac{1}{2} \left( \check{\alpha}_{i,x} + \check{\beta}_{i,x} \right)$$

is the Fenchel conjugate of $\ell_{i,x}$, and is defined for $-\check{\beta}_{i,x} \leq \xi_{i,x} \leq \check{\alpha}_{i,x}$ (these bounds become box constraints on the $\xi$s of Equation 19).

### F.2  Cutting plane $b$-optimization

Having described in the previous section how we may optimize over $w$ for fixed $b$ and $v$ using SDCA, we now move on to the problem of creating the SVMOptimizer needed by Algorithm 2, which must optimize over both $w$ and $b$.

Many linear SVM optimizers do not natively handle an unregularized bias parameter $b$, and this has long been recognized as a potential issue. For example, Shalev-Shwartz et al. [25] suggest using Pegasos to perform inner optimizations over $w$, and a bisection-based outer optimization over $b$. Our proposal is basically this, except that Algorithm 3, rather than using bisection, optimizes over $b$ using essentially the same cutting plane algorithm as we used in Algorithm 2, except that optimizing over $b$ is a minimization problem (over $v$ is maximization), and we might increase $u'_0$ on line 2 of Algorithm 3 for a technical reason (it will be needed by the proof of Lemma 6, but is probably not helpful in practice).

**Algorithm 3** Skeleton of a cutting-plane algorithm that finds a $b \in \mathcal{B}$ minimizing (to within $\epsilon$) $\min_{b \in \mathcal{B}, w \in \mathbb{R}^d} \Psi(w, b, v; w', b')$, where $\mathcal{B} \subseteq \mathbb{R}$ is a closed interval. It is assumed that $\tilde{u}'_0 \in \mathbb{R}$ is a finite upper bound on $\min_{b \in \mathcal{B}, w \in \mathbb{R}^d} \Psi(w, b, v; w', b')$, while by the definition of $\Psi$ (Equation 7), the $l'_0$ chosen on line 1 will lower bound the same quantity. The $u'_0$ increase that is "maybe" performed on line 2, and the CutChooser function on line 9, are discussed in Appendix F.2. The SDCAOptimizer function is as described in Appendix F.1.

---

SVMOptimizer $(v, \tilde{u}'_0, \epsilon')$

**1**    Initialize $g'_0 \in \mathbb{R}$ to zero and $l'_0 = -\sum_{j=1}^{m} v_j \check{\gamma}^{(j)}$

**2**    *Maybe* set $u'_0 = 2\tilde{u}'_0 - l'_0$, otherwise $u'_0 = \tilde{u}'_0$    *// needed for Lemma 6*

**3**    For $t \in \{1, 2, \dots\}$

**4**        Let $h'_t(b) = \max_{s \in \{0, 1, \dots, t-1\}} (l'_s + g'_s (b - b_s))$

**5**        Let $L'_t = \min_{b \in \mathcal{B}} h'_t(b)$ and $U'_t = \min_{s \in \{0, 1, \dots, t-1\}} u'_s$

**6**        If $U'_t - L'_t \le \epsilon'$ then

**7**            Let $s \in \{1, \dots, t-1\}$ be an index minimizing $u'_s$

**8**            Return $w^{(s)}, b_s, L'_t$

**9**        Let $b_t, \epsilon'_t = $ CutChooser $(h'_t, U'_t)$

**10**       Let $\xi^{(t)}, w^{(t)} = $ SDCAOptimizer $(b_t, v, \epsilon'_t)$

**11**       Let $u'_t = \Psi(w^{(t)}, b_t, v; w', b')$

**12**       Let $l'_t = \Psi^*(\xi^{(t)}, b_t, v; w', b')$ and $g'_t = \frac{\partial}{\partial'_b} \Psi^*(\xi^{(t)}, b_t, v; w', b')$

---

### F.2.1   Minimization-based

Perhaps the easiest-to-implement version of Algorithm 3 is based on the idea of simply solving for the minimizer of $h'_t$ at every iteration.

**Definition 3.** *(Minimization-based Algorithm 3)* Do not *increase $u'_0$ on line 2, and have CutChooser choose $b_t = \text{argmin}_{b \in \mathcal{B}} h'_t(b)$ and $\epsilon'_t = (U_t - L_t)/2$.*

As was the case in Appendix E.1, we have no convergence rates for this version. Furthermore, since this is a one-dimensional problem, the center of mass-based version of Algorithm 3 is implementable and efficient, so this minimization-based approach is not recommended.

### F.2.2   Center of mass-based

Essentially the same center of mass-based approach as was described in Appendix E.2 can be used in this setting, except that we must find the center of mass of a 2-dimensional sublevel epigraph, rather than an $m + 1$-dimensional superlevel hypograph:

**Definition 4.** *(Center of mass-based Algorithm 3)* Do *increase $u'_0$ on line 2, have CutChooser take $(b_t, z_t)$ to be the center of mass of $\{(b, z) \mid h'_t(b) \le z \le U'_t\}$, and choose $\epsilon'_t = (U'_t - z_t)/2$.*

Unlike in Appendix E.1, the fact that this problem is one-dimensional enables us to efficiently implement this CutChooser by explicitly representing each $h'_t$ as a set of piecewise linear segments, over which computing an integral (and therefore the center of mass) is straightforward, with a runtime that is linear in the number of segments.

Due to the similarity between Algorithms 3 and 2, we can simply recycle the results of Appendix E.2, with the troublesome second term in the bound of Lemma 4 removed by combining the "maybe" portion of Algorithm 3 with the Lipschitz continuity of $\Psi$ as a function of $b$:

**Lemma 6.** *In the context of Algorithm 3, suppose that we choose $b_t$ and $\epsilon'_t$ as in Definition 4. Then there is a iteration count $T_{\epsilon'}$ satisfying:*

$$T_{\epsilon'} = O\left(\ln\left(\frac{LB\left(\tilde{u}'_0 - l'_0\right)}{\epsilon'}\right)\right),$$

*such that, if $t \ge T_{\epsilon'}$, then $U'_t - L'_t \le \epsilon'$, where $B$ is the length of $\mathcal{B}$ and $L$ is as in Equation 18. Hence, Algorithm 3 will terminate after $T_{\epsilon'}$ iterations.*

*Proof.* Starting from (and adapting) the final equation in the proof of Lemma 4:

$$U_t' - L_t' \leq 4 \left(1 - \frac{1}{2e}\right)^{t-1} \left(\frac{1}{\epsilon'}\right) (u_0' - l_0')^2$$
$$\cdot \left(\frac{B}{\mu \left(\{b \in \mathcal{B} \mid \min_{w \in \mathbb{R}^d} \Psi(w, b, v; w', b') \leq u_0'\}\right)}\right).$$

Observe that, as a function of $b$, $\Psi(w, b, v; w', b')$ is $L$-Lipschitz. Hence, if we let $w^* \in \mathbb{R}^d, b^* \in \mathcal{B}$ be the optimal weight and bias, then:

$$\mu \left(\{b \in \mathcal{B} \mid \Psi(w^*, b, v; w', b') \leq u_0'\}\right) \geq \min \left\{B, \frac{u_0' - \Psi(w^*, b^*, v; w', b')}{L}\right\}.$$

Since $\min_{w \in \mathbb{R}^d} \Psi(w, b, v; w', b') \leq \Psi(w^*, b, v; w', b')$, it follows that:

$$U_t' - L_t' \leq 4 \left(1 - \frac{1}{2e}\right)^{t-1} \left(\frac{1}{\epsilon'}\right) (u_0' - l_0')^2 \max \left\{1, \frac{LB}{u_0' - \Psi(w^*, b^*, v; w', b')}\right\}.$$

This is the reason that we increased $u_0'$ on line 2 of Algorithm 3, since doing so has the result that $u_0' - \Psi(w^*, b^*, v; w', b') \geq \tilde{u}_0' - l_0'$. Since we also have that $u_0' - l_0' = 2(\tilde{u}_0' - l_0')$:

$$U_t' - L_t' \leq 16 \left(1 - \frac{1}{2e}\right)^{t-1} \left(\frac{1}{\epsilon'}\right) (\tilde{u}_0' - l_0') \max \{\tilde{u}_0' - l_0', LB\}.$$

The same reasoning as was used in the proof of Lemma 4 then gives the claimed bound on $T_{\epsilon'}$. $\square$

In addition to the above result, the obvious analogue of Lemma 5 holds as well:

**Lemma 7.** *In the context of Algorithm 3, suppose that we choose $b_t$ and $\epsilon_t'$ as in Definition 4. Then:*

$$\epsilon_t' \geq \frac{U_t' - L_t'}{2e},$$

*and in particular, for all $t$ (before termination):*

$$\epsilon_t' \geq \frac{\epsilon'}{2e},$$

*since we terminate as soon as $U_t' - L_t' \leq \epsilon'$.*

*Proof.* Same as Lemma 5. $\square$

In Appendix G, we'll combine these results with those of Appendices F.1 and E to bound the overall convergence rate of Algorithm 2.

### F.3 Kernelization

The foregoing discussion covers the case in which we wish to learn a linear classifier, and use an SVM optimizer (SDCA) that doesn't handle an unregularized bias. It's clear that we could freely substitute another linear SVM optimizer for SDCA, as long as it finds both a primal and dual solution so that we can calculate the lower and upper bounds required by Algorithm 2.

Our technique is easily kernelized—the resulting algorithm simply depends on inner kernel SVM optimizations, rather than linear SVM optimizations. SDCA can be used in the kernel setting, but the per-iteration cost increases from $O(d)$ arithmetic operations to $O(n)$ kernel evaluations, where $n$ is the total size of all of the datasets. Kernel-specific optimizers, such as LIBSVM [6], will generally work better than SDCA in practice, since they typically have the same per-iteration cost, but each iteration is "smarter". More importantly, such optimizers usually jointly optimize over $w$ and $b$, eliminating the need for Algorithm 3 entirely—in other words, these algorithms could be used to implement the higher-level SVMOptimizer, instead of the lower-level SDCAOptimizer. For this reason, an implementation based on such an optimizer is the simplest version of our proposed approach.

# G   Overall convergence rates

We may now combine the results in Appendices E and F into one bound on the overall convergence rate of Algorithm 2, assuming that we use Algorithm 3, rather than an off-the-shelf SVM solver, to implement the SVMOptimizer:

**Theorem 5.** *Suppose that we take $l_0 = -\sum_{j=1}^m v_j \gamma^{(j)}$ in Algorithm 2, that SVMOptimizer is implemented as in Algorithm 3, and that the CutChooser functions in Algorithms 2 and 3 are implemented using the center of mass (as in Definitions 2 and 4). Then Algorithm 2 will perform:*

$$O\left(m \ln\left(\frac{u_0 - l_0}{\epsilon}\right) + \ln\left(\frac{\mu(\mathcal{V})}{\mu(S_\ell(F, l_0))}\right)\right)$$

*iterations, each of which contains a single call to Algorithm 3, with each such call requiring:*

$$O\left(\ln\left(\frac{LBm(u_0 - l_0)}{\epsilon}\right)\right)$$

*iterations, each of which contains a single call to SDCAOptimizer, with each such call requiring:*

$$O\left(\max\left\{0, n \ln\left(\frac{\lambda n}{L^2 X^2}\right)\right\} + n + \frac{L^2 X^2 m}{\lambda \epsilon}\right)$$

*iterations, each of which requires $O(d)$ arithmetic operations.*

*Proof.* Notice that $u_0 \geq \tilde{u}_0' \geq l_0' \geq l_0$ (the first inequality because Algorithm 2 passes a quantity upper bounded by $u_0$ to SVMOptimizer, and the second by our choice of $l_0$). By Lemma 5, we also have that $\epsilon' \geq \epsilon/2e(m+1)$. The claimed results follow immediately from these facts, combined with Lemmas 4, 5, 6 and 7 and Theorem 4. $\qquad\square$

We can simplify (or perhaps *over*simplify) this result by considering only the total number of training examples $n$, number of constraints $m$, number of datasets $k$, dimension $d$ and desired suboptimality $\epsilon$, dropping all of the other factors, and assuming that the sizes of the $k$ datasets differ only by a constant factor (so that, as explained in Appendix F.1, we can take the Lipschitz constant $L$ to be of order $k$). Then the overall cost of finding an $\epsilon$-suboptimal solution to Problem 3 will be $\tilde{O}\left(dnm + dm^2 k^2/\epsilon\right)$ total arithmetic operations in the inner SDCA optimizers, plus $O\left(m \ln^2(k/\epsilon)\right)$ calls to the center of mass oracles in Algorithms 2 and 3, and another $O\left(m \ln^2(k/\epsilon)\right)$ calls to a linear programming oracle for finding $U_t$ in Algorithm 2 and $L_t$ in Algorithm 3.

We must reiterate that, as we mentioned in Appendix E.2, finding the center of mass is a computationally difficult problem. Hence, our reliance on a center of mass oracle for the optimization over $v$ is unrealistic (there is no problem when optimizing over $b$, since the underlying problem is one-dimensional). With that said, we hope that these results can provide a basis for future work.