[Reviews · NeurIPS 2016]

Reviewer 1

Summary

The paper tackles the problem of linear classification with realistic constraints of various types (churn, fairness, stability, recall, …) that can be modelled as dataset constraints of a particular structure. The focus of the paper is on the modelling part; the main optimization model, (2), is discontinuous, involves an L2 regularizer, a loss which is a positive linear combination prediction rates for the datasets considered, and a (small) number of constraints of a similar structure. The prediction rates are replaced by expected prediction rates (the classifier is made into a randomized one), which transform the main model into a continuous (but nonconvex) optimization problem. This problem is then solved using a Lagrangian approach (removing the constraints and transfering them to the objective with dual multipliers), coupled with an iterative majorization approach where the non-convex functions are replaced by convex upper bounds tight at the current iterate. It is suggested that a cutting plane approach be used for this. Experiments with fairness and churn problems show that the proposed approach works well in practice.

Qualitative Assessment

The paper is very well written, with a minimum number of typos and issues. The main problem is well motivated through considerations arising in practice, and hence there is very good potential for pick up by practitioners. It is interesting to see that a large number of practical goals can be modelled via dataset constraints of the type the authors consider. In some sense, the proposed methodology is straightforward given the basic idea – but seems to be rather of a tour-de-force nature, involving many elements (introduction of the ramp function to make the problem continuous, introducing Lagrange multipliers, iterative majorization by convex functions, cutting-plane technique + SDCA, …) . It is interesting to see that the authors have an open-source Julia implementation of the method. There is prior work which motivates the developments here, but nearly all of this work seems to be rather dated, and I think it is a good time to introduce similar problems to the attention of NIPS again. The main body of work does not contain any theory and is purely of a modelling / expository nature. A sizeable supplementary material is included which I have not used in this review (the authors deemed it not important enough to include it in the main text). Small issue: Figure 1 seems to be missing labels. -------- post-rebuttal remarks -------- I have read the rebuttal and keep my scores unchanged.

Confidence in this Review

2-Confident (read it all; understood it all reasonably well)


Reviewer 2

Summary

The paper proposes a model and algorithm to solve linear classification problems on datasets with some common side-constraints that are often ignored.

Qualitative Assessment

I really like the basic premise of the paper. I am no expert on deployed systems, but it seems like these authors know about real-world systems (and have access to some proprietary real datasets) and have thought carefully about actual issues that come up when making a classifier. The paper proposes (1) a new mindset, (2) a new model, and (3) an algorithm to solve the model. At times, I think any one of these components is a bit thin, and it’s not always clear why certain decisions (especially for the algorithm) were made — in particular, the numerical examples are provided to validate the new mindset but do not justify the algorithmic decisions. But overall, I think this is sufficiently novel that I am supportive. A few specific points: - I found the descriptions of the issues on page 1 and 2 (e.g., the churn example) quite interesting - Page 3 and further in the paper, referring to “Problem 2” is a bit confusing when it is only labeled by an equation number (most authors define a problem environment, like a theorem). I also didn’t see r_p and r_n defined (line 103) — are these s_p and s_n, or a (un)normalized version? - The randomized prediction (line 105) didn’t seem like it was fully explored. Does this affect training or only the final prediction? After vaguely introducing this concept, I felt like it was ignored until the numerical experiment plots, and I do not understand exactly what you are doing. - Fig 1 was designed to make the convex bounds intuitive, but this means that for z >= 1/2, at the next step, there is no actual dependence on this component of z? With the hinge loss, at least there is still dependence. Intuitively this makes your scheme sound like a bad idea. — Furthermore, discussing Davenport (line 178-180), you say that ramp-loss is similar or slightly better in accuracy than hinge-loss, but much faster. This doesn’t make sense to me, since to solve the ramp-loss you must solve a sequence of hinge-loss problems, so how can it be faster? Could you elaborate? - Section 2.2 uses SVM solvers inside a cutting-plane routine. This may be a good idea, but you could describe why you didn’t pursue alternatives. Everything is piecewise linear, except the regularization which is quadratic, so you could solve everything with an off-the-shelf QP solver, which seems much simpler. Perhaps you prefer existing SVM solvers since you know good ones, and many of them naturally take in datasets in specified formats (such as libsvm format)? A bit of discussion on your design decisions would be useful.

Confidence in this Review

2-Confident (read it all; understood it all reasonably well)


Reviewer 3

Summary

This paper proposes handling multiple goals on multiple datasets by training with dataset constraints, using the eamp penalty to accurately quantify costs, and present an efficient algorithm to approximately solve the resulting non-convex optimization problem. Experiments demonstrate the effectiveness of the proposed approach.

Qualitative Assessment

In my opinion, the novelty of the paper lies in integrating multiple objectives on multiple datasets, while traditional formulations are mainly one objective on one dataset. So conceptually, the paper is novel. However, only a simple linear classifier was learnt and the whole solution method was customized towards the goals that the authors listed in Introduction, making the proposed algorithm unuseful if one is interested in other goals. I would encourage the authors to discuss: 1. Is there a principled general solver that might work for general classifiers and general goals? 2. Multiobjective optimization can also work for multiple goals, but it does not combine the objectives by linear weighting. How is the proposed approach related to multiobjective optimization? 3. How to choose the weights for different goals? ****************** Comments after reading the authors' feedback: Although the author feedback is OK to me, I still feel that the paper is a bit strange. It is not in a traditional formulation. I would lean towards rejection. The reason is that: as I commented, the novelty is in integrating multiple objective across multiple datasets. So in this regard, the authors should elaborate more on why and how to integrate multiple objective across multiple datasets (e.g., the general solver and general classfier issues that I raised), and then give some examples, including the one shown in the paper, to illustrate the idea. However, the authors simply sell their sole example from the beginning. The whole paper and supplementary matrial is on defining the particular example and how to solve it in detail, which is not interesting to me at all.

Confidence in this Review

2-Confident (read it all; understood it all reasonably well)


Reviewer 4

Summary

In this paper, the authors try to combine misclassification error rate as well as other real-world goals, such as Churn, Coverage, Fairness, etc., into the classification problem by converting them as dataset constraints. The authors formulate the problem as an optimization problem. By replacing the indicator function with ramp function in the original optimization problem, the authors finally come up with an iterative algorithm to solve the problem.

Qualitative Assessment

1. The paper should be self-contained. Since part of the contribution in this paper is converting the real-world goals into dataset constraints, it is better to include the explanations of how to convert the goals into the constraints of the optimization problem in the paper instead of the supplemental material. 2. The complexity of the algorithm should be included in the paper as well so that the readers can have an idea about the scalability of the algorithm by just reading the paper. 3. There are some typos that need to be fixed.

Confidence in this Review

2-Confident (read it all; understood it all reasonably well)


Reviewer 5

Summary

This work aims at solving multiple goals, namely, coverage, churn, recall & precision, and egregious examples, using various datasets specific to these goals. This is achieved by introducing dataset constraints and proposing an algorithm to approximately optimize the non-convex problem.

Qualitative Assessment

I acknowledge the work done in tackling various possibilities in the goals and creating formulation for those types. In terms of organization of paper, it can be improved to make it more readable. I also recommend having a conclusion section in the end.

Confidence in this Review

1-Less confident (might not have understood significant parts)


Reviewer 6

Summary

This paper considers the problem of fitting a linear classifier to data, so that it minimizes/satisfies multiple objectives/constraints; the scenario in mind here is choosing a classifier for release to a real-world production system, where multiple (possibly competing) performance criteria must be achieved/satisfied. This is a useful goal, and one that has not received a ton of attention in the literature so far. The authors cast this as an optimization problem, present a relaxation of this problem, and then derive a (majorization/minimization) algorithm for optimizing the relaxation. Two numerical studies are presented.

Qualitative Assessment

In addition to my above comments re: empirical evaluation, here are some further remarks. Approach: - Why not simply use the hinge (or squared) loss relaxation(s) of the 0/1 loss (which are convex) instead of using the ramp loss? This would make for a much simpler and possibly faster optimization approach, since it appears that scalability may be an issue you are concerned with. - As you mention (L183), it looks like Joachims (2005) previously considered using the hinge loss as a relaxation (at least for a subset of the performance criteria you care about). I think, to demonstrate the efficacy of your method, you need to compare your approach to his (at least on some criteria like, say, scalability); otherwise it's not clear what the value-add of your approach is. - L144-150: I'm a little confused, I thought most SVM solvers return the primal and dual variables ... why do you need an outer cutting plane method to compute the dual variables? - How many iterations does it typically take for your majorization/minimization (MM) approach take to converge? Do you have any timing results? It sounds like your approach may be costly.

Confidence in this Review

2-Confident (read it all; understood it all reasonably well)